# Chalcone-Acridine Hybrid Suppresses Melanoma Cell Progression via G2/M Cell Cycle Arrest, DNA Damage, Apoptosis, and Modulation of MAP Kinases Activity

**DOI:** 10.3390/ijms232012266

**Published:** 2022-10-14

**Authors:** Maria Gazdova, Radka Michalkova, Martin Kello, Maria Vilkova, Zuzana Kudlickova, Janette Baloghova, Ladislav Mirossay, Jan Mojzis

**Affiliations:** 1Department of Pharmacology, Faculty of Medicine, Pavol Jozef Šafárik University, 040 01 Košice, Slovakia; 2NMR Laboratory, Institute of Chemistry, Faculty of Science, Pavol Jozef Šafárik University, 040 01 Košice, Slovakia; 3Department of Dermatovenerology, Faculty of Medicine, Pavol Jozef Šafárik University, 040 01 Košice, Slovakia

**Keywords:** chalcone-acridine hybrid, apoptosis, cell cycle arrest, DNA damage, MAP kinases

## Abstract

This study was focused on investigating the antiproliferative effects of chalcone hybrids in melanoma cancer cells. Among seven chalcone hybrids, the chalcone-acridine hybrid **1C** was the most potent and was selected for further antiproliferative mechanism studies. This in vitro study revealed the potent antiproliferative effect of **1C** via cell cycle arrest and apoptosis induction. Cell cycle arrest at the G2/M phase was associated with modulation of expression or phosphorylation of specific cell cycle-associated proteins (cyclin B1, p21, and ChK1), tubulins, as well as with the activation of the DNA damage response pathway. Chalcone **1C** also induced apoptosis accompanied by mitochondrial dysfunction evidenced by a decrease in mitochondrial membrane potential, increase in Bax/Bcl-xL ratio and cytochrome c release followed by caspase 3/7 activation. In addition, increased phosphorylation of MAP kinases (Erk1/2, p38 and JNK) was observed in chalcone **1C**-treated melanoma cells. The strong antiproliferative activities of this chalcone-acridine hybrid suggest that it may be useful as an antimelanoma agent in humans.

## 1. Introduction

The incidence of melanoma worldwide is increasing at a greater rate than other types of cancer (1.7% of global cancer diagnoses) [1]. The prevalence of skin melanoma differs among populations. The disease occurs mostly in white-skinned Caucasian populations in Australia and New Zealand [2]. In Europe, melanoma annually claims more than 20,000 lives, and it is a significant public health burden [3]. The therapy of melanoma depends on the stage of disease but generally includes surgical excision, treatment with immune checkpoint inhibitors, targeted therapy, radiotherapy or chemotherapy [4]. Although mortality rates have fallen over the past decade with the approval of new targeted therapies such as BRAF and MEK inhibitors and immune checkpoint inhibitors, many of the current anti-melanoma drugs are expensive and toxic [5,6,7]. Furthermore, despite the substantial therapeutic outcome of modern therapy, some patients acquire drug resistance and melanoma recurrence. For this reason, the development of novel, less toxic therapeutics for patients with melanoma remains essential.

In the last few years, plant-derived natural compounds have been extensively studied for their anti-proliferative and anti-cancer effects [8,9,10,11,12,13]. Phytochemicals are predominantly attractive because of their availability, low toxicity and absence of serious adverse reactions [14,15]. Furthermore, several clinical trials have reported that natural compounds tested either as single agent or in combination with standard chemotherapeutic drugs improve sensitivity to chemotherapy and radiotherapy along with the survival of patients [16,17,18,19,20]. 

Chalcones, the precursors of flavonoids and isoflavonoids in plants, have been shown to display a broad spectrum of biological actions including anti-inflammatory [21], antioxidant [22], immunomodulatory [23], antidiabetic [24], antibacterial [25,26,27], antiviral, [28] and antiparasitic [29] effects. Furthermore, the anticancer, chemopreventive and antiangiogenic activities of chalcones have also been documented [30,31,32,33].

Due to the structural heterogeneity, chalcones are useful templates for the development of novel active compounds with more convenient biological activities [34,35]. In the last decade we documented the antiproliferative effect of several chalcone derivatives such as acridine hybrids [36,37,38], indole hybrids [39,40,41] or cyclic chalcone analogues [42,43,44] using different in vitro cancer models such as breast, colorectal or cervix cancers. Furthermore, both the antiproliferative and anticancer effects of chalcones have also been documented using melanoma cancer cells or melanoma xenografts [45,46,47,48,49,50,51,52,53]. 

In the present study, we investigated the mechanism of acridine chalcone **1C** on the induction of apoptosis in A2058 and BLM melanoma cells. Our results indicate that the antiproliferative effect of chalcone **1C** is associated with the induction of an intrinsic pathway of apoptosis, G2/M cell cycle arrest, DNA damage and the modulation of selected signaling pathways. To the best of our knowledge, this is the first study displaying the antiproliferative activity of the chalcone-acridine hybrid against melanoma cancer cell lines.

## 2. Results

### 2.1. MTT Screening Assay

The effect of synthetic chalcone derivatives on the metabolic activity of selected melanoma and healthy cell lines was determined using the MTT assay. Chalcone derivates suppressed cell metabolism with IC50 values ranging from 7.96 to ˃100 μmol/L (Table 1). For further experiments, the most potent acridine chalcone **1C** was selected using a concentration of **1C** 10 μmol/L for A2058 and 20 μmol/L for the BLM melanoma cell line. DMSO was used as negative control with no effect on melanoma cell growth.

### 2.2. BrdU Cell Proliferation Assay

The BrdU Cell Proliferation Assay is based on the detection of BrdU (5-bromo-2′-deoxyuridine) incorporated into the newly synthesized DNA during the replication process in proliferating cells. The results showed that chalcone **1C** suppressed proliferation of melanoma cells ranging from 5 to 20 μmol/L compared with DMSO control (Figure 1). The IC50 values were 8.40 ± 0.05 μmol/L (A2058) and 16.51 ± 0.01 μmol/L (BLM) after 72 h of incubation. The comparison of IC50 from MTT and BrdU assay is shown in Table 2.

### 2.3. AO/PI Apoptosis Analysis

Staining with acridine orange (AO) and propidium iodide (PI) is a method for dividing cells into populations according to whether they are living, apoptotic or necrotic. AO is a dye binding to living and dead cells while PI only stains cells with lost membrane integrity. As shown in Figure 2, the chalcone **1C** caused a significant decrease in proliferation and viability of A2058 and BLM melanoma cells. The number of apoptotic (yellow and orange) cells increased with a time-dependent trend simultaneously with the increased detachment of cells as a result of lost adhesion. The results of AO/PI staining, annexin V/PI staining and cell cycle analysis show that chalcone **1C** induces the apoptotic cell death of melanoma cells.

### 2.4. Cell Cycle Analysis

To determine whether cell cycle arrest is related to the inhibition of cell proliferation by chalcone **1C** in A2058 and BLM melanoma cells, the cell cycle progression was examined by flow cytometry analysis. As demonstrated in Table 3 and Figure 3, treatment with chalcone **1C** increased the number of melanoma cells in the G2/M phase at all exposition times (24, 48 and 72 h). Furthermore, a significant increase in the subG0 population (the marker of apoptosis) was observed in both melanoma cell lines with the highest peak after 72 h of treatment with chalcone **1C**. The results suggest that the antiproliferative effect of chalcone **1C** in A2058 and BLM melanoma cells can be associated with G2/M cell cycle arrest and the induction of apoptosis in a time-dependent pattern.

### 2.5. Apoptosis Detection

Phosphatidylserine (PS) is an anionic phospholipid located on the cytoplasmic surface of the plasma membrane in cells with intact membranes. The externalization of PS from the inner to outer surface of the lipid bilayer of plasma membrane is a typical marker of apoptosis in the early stages. Annexin V/PI double staining is a method for dividing cells into populations of living (An^−^/PI^−^ Q3), early apoptotic (An^+^/PI^−^ Q4), late apoptotic/necrotic (An^+^/PI^+^ Q2) and dead (An^−^/PI^+^ Q1). The analysis showed the significant increase of early apoptotic and late apoptotic/necrotic A2058 and BLM melanoma cells after 48 and 72 h of treatment with chalcone **1C**. Furthermore, the concomitant decrease of living cells and the increased number of dead cells were also observed in a time-dependent pattern (Table 4, Figure 4).

### 2.6. Effect of Chalcone **1C** on Cell Cycle-Related Protein Expression

#### 2.6.1. p53 Protein

The p53 is an important tumor suppressor protein activated when cells experience stress stimuli such as DNA damage and other genetic alterations. Phosphorylation of p53 protein disrupts the binding with its negative regulator Mdm2, which results in either cell cycle arrest, DNA repair or apoptosis. WB analysis showed significant increase in the amount of total and phosphorylated (maximum at 72 h) form of p53 protein in A2058 melanoma cells after treatment with chalcone **1C** (Figure 5, Appendix A). In BLM melanoma cells the total p53 was maintenanted on basic levels as DMSO treated controls, with little decrease at 24 h. In comparison with A2058, in BLM cells phosphorylation increased late on 72 h.

#### 2.6.2. p21 Protein

The p21Waf1/Cip1 is a tumor suppressor protein involved in cell cycle regulation. The inhibition of cyclin-CDK complexes by p21 protein leads to the blockade of cell cycle progression in the G1/S and G2/M phase. The increased expression of p21 protein is subsequently followed by phosphorylation of p53 protein as a response to DNA damage. A western blot analysis revealed a significant increase in the expression of p21 protein in A2058 melanoma cells after treatment with chalcone **1C** at concentration 10 μmol/L after 24 and 48 h. In contrast with A2058 cells, BLM melanoma cells showed an increase and maintained a trend in the expression of p21 protein after treatment with tested chalcone at concentrations of 20 μmol/L (Figure 6, Appendix A).

#### 2.6.3. Chk1 Kinase

Chk1 is a serine/threonine kinase involved in cellular response to DNA damage and activation of cell cycle checkpoints. Chk1 is phosphorylated by activated ATR kinase and is restricted to S and G2 phases. Results showed that chalcone **1C** significantly induced the phosphorylation of Chk1 in A2058 melanoma cells after 24, 48 (maximum) and 72 h of treatment. Analysis confirmed a consilient trend of Chk1 phosphorylation in BLM melanoma cells (Figure 6, Appendix A).

#### 2.6.4. Cyclin B1

Cyclin B1 is a regulatory protein expressed predominantly during the G2/M phase. The activated cyclin B1-Cdk1 complex is involved in early events of mitosis, such as chromosome condensation, breakdown of the nuclear envelope and the assembly of the spindle apparatus. The analysis showed significant downregulation of phosphorylated cyclin B1 in A2058 and BLM melanoma cells after treatment with chalcone **1C** with maximum after 72 h in both cell lines (Figure 6, Appendix A).

#### 2.6.5. Effect on Tubulins

Microtubules, key components of the cytoskeleton, are polymers of α- and β-tubulins responsible for many cellular processes including cell division, cell migration and maintenance of cell structure. The dysregulation of tubulins can lead to G2/M cell cycle arrest or apoptosis. Western blot analysis revealed significant alterations in expression of α-, α**1C**- and β-tubulin in A2058 and BLM melanoma cells after treatment with chalcone **1C**. As shown in Figure 6 and Appendix A, the significant upregulation of α-, α**1C**- and β-tubulin was observed in the A2058 cell line with a maximum after 48 and 72 h. In the BLM cell line, results showed significant downregulation of α- and β-tubulin in a time-dependent pattern and the upregulation of α**1C**- tubulin with a maximum after 72 h of treatment.

### 2.7. Effect of Chalcone **1C** on Mitochondrial Apoptosis Pathway Proteins

#### 2.7.1. Effect on Bcl-2 Family Proteins

The Bcl-2 family proteins regulate the mitochondrial pathway of apoptosis. Some members of the family are involved in the inhibition of cell death (Bcl-2 and Bcl-xL), while others promote apoptosis (Bax and Bak). Proapoptotic proteins promote the permeabilization of mitochondrial membrane, the release of cytochrome c and inhibit the function of antiapoptotic proteins. As results showed, treatment with chalcone **1C** significantly increased amounts of proapoptotic protein Bax and Bad in A2058 and BLM melanoma cells with maximums at 48 h (Figure 7, Appendix A). Moreover, the phosphorylation of Bad increasing up to 48 h in both tested cell lines was observed. On the other hand, decreased levels of antiapoptotic protein Bcl-xL was observed in both cell lines. The total Bcl-2 protein was upregulated only in A2058 cells after **1C** treatment up to 48 h, while in BLM cells it was not. However, the phosphorylation of Bcl-2 increased in A2058 up to 48 h and decreased at 72 h, while in BLM it increased up to the 72 h time point after **1C** treatment (Figure 7, Appendix A).

#### 2.7.2. Cytochrome c Release

Cytochrome c is essential for controlling the energy metabolism of cells and apoptosis. As a result of apoptotic stimulus, cytochrome c is released from the mitochondria into cytosol, which activates programmed cell death. Results showed the significant release of cytochrome c in A2058 and BLM melanoma cells after 24, 48 and 72 h of treatment with chalcone **1C** (Figure 8).

#### 2.7.3. Mitochondrial Membrane Potential (MMP)

The mitochondria play a crucial role in the regulation of cell survival and the induction of apoptotic cell death. One of the key indicators of mitochondrial activity is mitochondrial membrane potential (MMP). The loss of MMP due to mitochondrial dysfunction may lead to apoptosis. As shown in Figure 9, the treatment of A2058 and BLM melanoma cells with chalcone **1C** caused a significant increase in the percentage of cells with reduced MMP at all exposition times (24, 48 and 72 h).

#### 2.7.4. Caspase 3/7 Activity

Caspases are enzymes with specific cysteine protease activity involved in programmed cell death. As a result of mitochondrial dysfunction, cytochrome c is released into cytosol with the subsequent formation of apoptosome and the activation of caspases. A flow cytometric analysis showed that treatment of A2058 and BLM melanoma cells with chalcone **1C** caused significant activity of executioner caspase-3/7 in apoptotic cells (death cells excluded) after 24, 48 and 72 h (Figure 10).

#### 2.7.5. Cleavage of PARP

Poly (ADP-ribose) polymerase (PARP) plays an important role in DNA repair and programmed cell death. The activation of caspases leads to cleavage of PARP resulting in DNA repair inhibition. As shown in Figure 11 and Appendix A, the levels of cleaved PARP (89 kDa) significantly increased in A2058 and BLM melanoma cells after treatment with chalcone **1C** at all exposition times (24, 48 and 72 h).

### 2.8. DNA Damage Analysis

#### 2.8.1. ATM Kinase

The ATM (ataxia-telangiectasia mutated) serine/threonine kinase phosphorylates several key proteins that regulate DNA repair, cell cycle checkpoint control and apoptosis. The phosphorylation of ATM is activated by DNA double-strand breaks. As demonstrated in Figure 12A,B, chalcone **1C** significantly induced phosphorylation of ATM in A2058 and BLM melanoma cells in a time-dependent manner (24, 48 and 72 h).

#### 2.8.2. SMC1 Protein

The SMC1 (structural maintenance of chromosomes 1) protein is a member of key proteins regulating DNA repair and the cohesion of chromosomes during the cell cycle. SMC1 activation is mediated by ATM as a response to DNA damage. Analysis showed the significant phosphorylation of SMC1 in A2058 and BLM melanoma cells after 24, 48 and 72 h of treatment with chalcone **1C** (Figure 12C,D).

#### 2.8.3. Histone HA2.X

The Histone H2A.X is a histone variant essential for DNA repair. The phosphorylation of H2A.X occurs due to double-strand DNA breaks induced by genotoxic stress resulting in DNA repair, cell cycle arrest or apoptosis. Treatment of A2058 and BLM melanoma cells with chalcone **1C** significantly induced phosphorylation of H2A.X after 24, 48 and 72 h (Figure 12E,F).

### 2.9. Chalcone **1C** Modulates Signalling Pathways/Changes in Expression and Phosphorylation of MAPK Proteins

Mitogen-activated protein kinases (MAPKs) are a family of proteins which modulate a series of vital signalling pathways involved in the regulation of cell proliferation, differentiation, survival and apoptosis. Each signalling pathway is initiated by external stimuli and leads to the activation of particular MAPK. The mammalian MAPKs are grouped into three families including extracellular signal-regulated kinases (ERKs), c-jun N-terminal kinases (JNKs) and p38s. Results from a WB analysis revealed that treatment of A2058 melanoma cells with chalcone **1C** significantly increased the phosphorylation of ERK1/2 with a maximum after 48 h. Analysis also confirmed a consilient trend of ERK1/2 phosphorylation in BLM melanoma cells. In both melanoma cell lines, chalcone **1C** significantly increased the phosphorylated form of p38 MAPK mainly after 48 h of incubation and phospho-JNK in all tested timepoints (Figure 13, Appendix A).

## 3. Discussion

As was mentioned above, numerous pharmacological activities (including antiproliferative and anticancer) have effects that have been attributed to chalcones [54,55]. It is well known that chalcone moiety is an effective template for the discovery of new anticancer drugs and its hybridization with other anticancer agents represents a promising approach to develop novel agents with high anticancer activity [56].

Acridine-based agents represent a family of heterocyclic compounds which are currently undergoing significant research because of their potential anticancer activity. The pleiotropic mechanism of the antiproliferative effect of acridine derivatives, including topoisomerase II inhibition [57], cell cycle arrest in S phase [58] and G2/M phase [59], tubulin polymerization inhibition [60], DNA damage [61] and apoptosis induction [58] have been documented.

On the other hand, the antiproliferative effect of chalcone-acridine hybrids has been studied only minimally. Recently, we described the antiproliferative effect of the chalcone-acridine hybrid in colorectal cancer cells [36]. At micromolar concentrations, this chalcone inhibited cell proliferation associated with G2/M block, and dysregulated of tubulin expression, apoptosis induction as well as the modulation of several signalling pathways associated with cell life and death. We later showed that the antiproliferative effect of this acridine chalcone is closely related to the generation of reactive oxygen species [37]. Moreover, it may overcome drug resistance as we observed growth suppression in P-glycoprotein-expressing cancer cells [62]. Furthermore, in the most recent study we documented the proapoptotic effect of the novel chalcone-acridine hybrid in breast cancer cells. In addition to the induction of mitochondrial apoptosis, DNA studies demonstrated that it interacts with DNA through bimodal binding mode, i.e., intercalation and groove-binding [38]. Besides this, some in silico studies also suggest the anticancer potential of chalcone-acridine hybrids [63,64]. 

In the present work, we studied possible mechanisms of action of either acridine or indole chalcone derivatives in vitro using a melanoma cancer model. Among the tested chalcones, a chalcone-acridine hybrid (**1C**) showed the highest antiproliferative potency and was selected for the next study.

In our recent article [41], we mentioned pleiotropic mechanisms of the antiproliferative effect of chalcones in in vitro cancer models. Among others, chalcones caused cell cycle arrest, mostly at the G2/M phase [36,65,66]. In the present paper, the exposure of melanoma cells to chalcone **1C** caused cell cycle arrest at the G2/M phase with simultaneous increase in cell number with sub-G0/G1 DNA content. Moreover, chalcone **1C** also affected the expression as well as the phosphorylation of specific cell cycle-associated proteins including cyclin B1, p21, and Chk1. Our results indicated that the suppression of cell proliferation and arrest at the G2/M phase of cell cycle in chalcone **1C**-treated melanoma cells may be related to the modulation of cycle-associated protein activity.

Furthermore, cell cycle arrest at the G2/M phase is the consequence of DNA damage and it is the last chance for cell repair prior to entering mitosis [67]. In response to DNA damage, the DNA damage response (DDR) pathway is activated [68], and the main components of DDR, ATM and ATR (ATM and Rad3-related) kinases subsequently phosphorylate several components involved in the cell cycle, DNA replication, DNA repair or apoptosis [69]. Our results showed that chalcone **1C**-induced a significant increase in ATM phosphorylation followed by its downstream molecules phosphorylation including histone H2A.X at Ser139 (γ-H2A.X), p53 and p21 proteins, SMC1, indicating that this chalcone has DNA toxicity. Recently, different chalcone hybrids have been reported to have an antiproliferative effect associated with DNA damage [48,70]. In addition, DDR is also associated with the activation of PARP, an enzyme involved in DNA repair [71]. Our experiments showed that chalcone **1C** induced the cleavage of PARP. After cleavage, PARP loses its function, resulting in the suppression of DNA repair [72]. 

Irreparable DNA damage is often associated with the induction of cell death, including apoptosis [73]. As was mentioned above, chalcone **1C** induced the increase in cell numbers with sub-G0/G1 DNA content, which is the result of internucleosomal DNA fragmentation and is considered a marker of apoptosis. This result prompted us to (i) confirm apoptosis, and (ii) study molecular mechanisms of **1C**-induced apoptosis in melanoma cells.

The translocation of PS from the cytoplasmic surface of the plasma membrane on the outer leaflet of the plasma membrane is one of the first events of apoptosis. Once externalized, PS can be visualised by annexin V staining [74]. In the present paper, we observed the significant increase in the number of apoptotic cells after 48 and 72 h of treatment in A2058 cells. In BLM melanoma cells, chalcone **1C** increased the number of apoptotic cells after 24–72 h of incubation. Furthermore, apoptosis has also been proven by acridine orange and propidium iodide staining.

Today, the mechanisms of the apoptosis-inducing effect of **1C** in melanoma cancer cells is not known. In the present study, we tried to discover the mechanism by which this chalcone induces apoptosis.

Mitochondria play a key role in cell life and death. Mitochondrial membrane potential (ΔΨm, MMP) is critical for maintaining the mitochondrial physiological function and its loss is often considered as an early event in apoptosis [75]. The decrease in MMP is closely related to the permeabilization of the mitochondrial outer membrane with the subsequent release of several proapoptotic proteins including cytochrome c, Smac/DIABLO or apoptosis-inducing factor [76]. In our study, chalcone **1C** significantly increased the number of cells with dissipated MMP after 24, 48 and 72 h of incubation and, concurrently, increased the cytosolic concentration of cytochrome c in both melanoma cell lines. Once cytochrome c is released, it interacts with apoptosis protease-activating factor 1, forming apoptosome, which mediates the activation of an initiator caspase-9 [77] followed by the activation of its downstream executioner caspases-3/7 [78]. In this context, our results showed the significant activation of caspases 3/7 in both melanoma cell lines after 24–72 h of incubation.

In addition, the integrity of the mitochondrial membrane is strictly regulated by the members of the Bcl-2 protein family [79]. The presented results showed the significant effect of chalcone **1C** on different members of these proteins. We found that the exposition of both melanoma cell lines to chalcone **1C** led to upregulation of the proapoptotic Bax protein and the downregulation of the antiapoptotic Bcl-xL protein. It has been documented that an increased Bax/Bcl-xL ratio supports apoptosis due to cytochrome c release with subsequent activation of caspases, as mentioned earlier [80]. Similar results were obtained with either synthetic or natural chalcones in different cancer cells [41,81]. Furthermore, we observed the increased phosphorylation of antiapoptotic Bcl-2 and proapoptotic Bad proteins. It is well known that the phosphorylation of the Bcl-2 family proteins is a key regulator of its function [82,83]. However, phosphorylation can be modulated by several factors, such as kinases involved in phosphorylation or target sites on Bcl-2 proteins resulting to either proapoptotic or antiapoptotic activity [84]. Several lines of evidence indicate that Bcl-2 phosphorylation induced by microtubule damaging agents led to the inactivation of its antiapoptotic function [85,86,87]. Furthermore, it has been documented that Bad phosphorylation is associated with the loss of its proapoptotic activity [88,89]. On the other hand, several experimental works showed that microtubule damaging agents-induced Bad phosphorylation promoted apoptosis [90,91]. As mentioned above, chalcone **1C** induced cell arrest at the G2/M phase of the cell cycle and simultaneously modulated tubulin expression, which indicated its potentially microtubule damaging activity. Recently, Liu and co-workers [92] have reviewed the ability of several chemically different chalcones to interfere with tubulin and to disturb the dynamic balance of microtubules. On the basis of the above-mentioned facts, we suggest that the modulation of the expression/phosphorylation of the Bcl-2 protein family is involved in **1C**-induced apoptosis in melanoma cells.

It is known that several protein kinases are involved in cell survival and death. Among them, MAPK is a key pathway related to apoptosis [93]. Of the main components of MAPK, ERK1/2 is mostly involved in cell survival, while the phosphorylation of JNK and p38 promote apoptosis [94]. 

A large number of studies have demonstrated that JNK and p38 phosphorylation is associated with apoptosis induced by various compounds such as taxanes [95], vinblastine [96] or doxorubicin [97]. In the present study, we documented increased JNK and p38 phosphorylation in both **1C**-treated melanoma cells. Our results agreed with previous studies where the association between JNK and p38 phosphorylation and chalcone-induced apoptosis was documented [40,41,98,99]. Surprisingly, we also found the activation (i.e., phosphorylation) of Erk1/2 which, generally, prevents the apoptosis via either downregulation of proapoptotic or the upregulation of antiapoptotic proteins [93]. On the other hand, the opposite activity has also been documented [100]. The activation of Erk1/2 may lead to mitochondrial membrane disruption with subsequent cytochrome c release [101], modulation of Bcl-2 family protein expression [102], or the suppression of the PI3K/Akt pathway [103]. Furthermore, Erk phosophorylation in anticancer drug-induced apoptosis has also been noted [104,105].

## 4. Materials and Methods

### 4.1. Tested Compounds

(2E)-3-(acridin-9-yl)-1-(2,6 dimethoxyphenyl)prop-2-en-1-one (**1C**, Figure 14), (2E)-1-(2-fluorophenyl)-3-(1-methoxy-1H-indol-3-yl)prop-2-en-1-one (ZKCH-11A), (2E)-3-(2-ethoxy-1H-indol-3-yl)-1-(2-fluorophenyl)prop-2-en-1-one (ZKCH-11C), (2E)-1-(2-fluorophenyl)-3-[2-(propan-2-yloxy)-1H-indol-3-yl]prop-2-en-1-one (ZKCH-11E), (2E)-3-(2-butoxy-1H-indol-3-yl)-1-(2-fluorophenyl)prop-2-en-1-one (ZKCH-11F), (2E)-1-(2-fluorophenyl)-3-[2-(2-methylpropoxy)-1H-indol-3-yl]prop-2-en-1-one (ZKCH-11G) and (2E)-1-(2-fluorophenyl)-3-(1H-indol-3-yl)prop-2-en-1-one (ZKCH-11H) were synthesized by Maria Vilkova (Faculty of Science, P.J. Šafárik University, Košice) and Zuzana Kudlickova (NMR Laboratory, Institute of Chemistry, Faculty of Science, P.J. Šafárik University, Košice). The structure of compounds was confirmed by using ^1^H, ^13^C nuclear magnetic resonance (NMR), infrared (IR) spectroscopy and mass spectrometry (MS), with 97% purity based quantitative NMR [106]. The studied compounds were dissolved in dimethyl sulfoxide (DMSO) with the final concentration of <0.2% in the culture medium. DMSO exhibited no cytotoxicity on cultured cells.

### 4.2. Cell Culture

Cell lines A2058 (human melanoma lymph node metastasis, cat. 91100402) from ECACC (Public Health England, Salisbury, UK) and BLM (human melanoma lung metastasis a gift from prof. K. Smetana, Institute of Anatomy, Charles University in Prague) were cultured in a medium consisting of high glucose Dulbecco’s Modified Eagle’s Medium (DMEM) and sodium pyruvate (GE Healthcare, Piscataway, NJ, USA). The growth medium was supplemented with a 10% fetal bovine serum (FBS) and antibiotic/antimycotic solution 1 × HyClone™ (GE Healthcare, Chicago, IL, USA). The MCF-10A (human mammary epithelial cells) cell line was cultured in a medium consisting of high glucose Dulbecco’s Modified Eagle’s Medium F12 (DMEM/F12) (Biosera, Kansas City, MO, USA). The growth medium was supplemented with a 10% FBS, antibiotic/antimycotic solution 1 × HyClone™ (GE Healthcare, Chicago, IL, USA), epidermal growth factor (EGF) (20 ng/mL final), hydrocortisone (0.5 μg/mL final) and insulin (10 μg/mL final) (Sigma-Aldrich Chemie, Steinheim, Germany). Cells were cultured in humidified air at 37 °C with atmosphere containing 5% CO_2_.

### 4.3. MTT Viability Assay

The half-maximal inhibitory concentration values (IC50) and metabolism inhibition of tested synthetic chalcone derivatives were determined by MTT (3-(4,5-di-methylthiazol-2-yl)2,5-diphenyltetrazolium bromide) colorimetric assay (Sigma-Aldrich Chemie, Steinheim, Germany). Tested cell lines were seeded at a density of 5 × 10^3^ cells/well in 96-well culture plates. After 24 h, the tested chalcones in concentrations of 100, 50 and 10 μmol/L were added and incubation proceeded for the next 72 h. In the next step, 10 μL of MTT (5 mg/mL) was added to each well containing cells and incubated for another 4 h at 37 °C during which MTT was metabolized to insoluble formazan in cells. After 4 h, 100 μL of a 10% sodium dodecyl sulphate (SDS) was added to each well and another 24 h were allowed for the formazan crystals to dissolve. The metabolic activity of cells was evaluated by measuring the absorbance at wavelength 540 nm using the automated Cytation™ 3 Cell Imaging Multi-Mode Reader (Biotek, Winooski, VT, USA). Three independent analyses were performed.

### 4.4. BrdU (5-Bromo-2′-deoxyuridine) Cell Proliferation Assay

The A2058 (5 × 10^3^/well) and BLM (5 × 10^3^/well) melanoma cells were plated in a 96-well culture plate in 80 μL suitable medium. Twenty-four hours after cell seeding, different concentrations of the chalcone **1C** were added ranging from 5–20 μmol/L. After 48 h of treatment, BrdU labelling solution was added to melanoma cells and incubated for another 24 h at 37 °C followed by fixation and incubation with anti-BrdU peroxidase conjugate solution for an additional 90 min in the dark at room temperature. Cells were then washed 3× with washing solution and incubated with TMB (tetramethylbenzidine) substrate solution (all Roche, Basel, Switzerland) for 5 to 30 min according to colour intensity. Finally, the stop solution (25 μL 1 M H_2_SO_4_) was added, and the incorporated BrdU was detected with an automated Cytation™ 3 Cell Imaging Multi-Mode Reader (Biotek, Winooski, VT, USA) at 450 nm. Three independent analyses were performed.

### 4.5. AO/PI Viability Assay

The A2058 and BLM melanoma cells were seeded at a density of 5 × 10^4^/well into 6-well culture plates. Twenty-four hours after seeding, A2058 cells were treated with chalcone **1C** at 10 μmol/L concentration, BLM cells at 20 μmol/L concentration and both melanoma cell lines were treated with vehicle (DMSO) at the same concentrations. At 24, 48 and 72 h after treatment, the culture medium was removed, the cells were washed with washing buffer (PBS) and fixed with 4% paraformaldehyde (pH 7.2) for 30 min. In the next step, the paraformaldehyde was removed, the cells were washed with PBS and the staining solution (10 μg/mL acridine orange and 10 μg/mL propidium iodide, Sigma-Aldrich) was added to each well for 1 h incubation at room temperature in the dark. Finally, the staining solution was removed, the cells were washed with PBS and apoptosis was observed using an automated Cytation™ 3 Cell Imaging Multi-Mode Reader (Biotek, Winooski, VT, USA).

### 4.6. Cell Cycle Analysis

The A2058 and BLM melanoma cells were seeded at a density of 1 × 10^5^/dish in a Petri dish. Twenty-four hours after incubation, A2058 cells were treated with chalcone **1C** at 10 μmol/L concentration, BLM cells at 20 μmol/L concentration and both melanoma cell lines were treated with DMSO as the negative control at the same concentrations. For flow cytometric analysis of the cell cycle, adherent and floating A2058 and BLM melanoma cells were harvested in three different times (24, 48 and 72 h) after treatment with chalcone **1C** and DMSO, washed in cold washing buffer (PBS), fixed in cold 70% ethanol and stored at −20 °C at least overnight. Prior to analysis, cells were washed with PBS, resuspended in staining solution (Triton X-100 final concentration 0.1%, ribonuclease A final concentration 0.5 mg/mL and propidium iodide final concentration 0.025 mg/mL, all Sigma), and incubated for 30 min in the dark at room temperature. Stained melanoma cells were analysed using a BD FACSCalibur^TM^ Flow Cytometer (Becton Dickinson, San Jose, CA, USA). Three independent analyses were performed.

### 4.7. Apoptosis Detection

To perform apoptosis detection, A2058 and BLM melanoma cells were seeded in Petri dishes at a density of 1 × 10^5^/dish and treated with chalcone **1C** and DMSO as the negative control at the same concentrations for 24, 48 and 72 h. In the next step, adherent and floating melanoma cells were harvested, centrifuged and pellets were resuspended in washing buffer (PBS). Resuspended melanoma cells were stained with Annexin V-Alexa Fluor^®^ 647 antibody (Thermo Scientific, Rockford, IL, USA) in binding buffer for 20 min in the dark at room temperature. Finally, melanoma cells were washed, stained with 1 μL of propidium iodide (final concentration 25 μg/mL) for 5 min and analysed using a BD FACSCalibur^TM^ Flow Cytometer (Becton Dickinson, San Jose, CA, USA). Three independent analyses were performed.

### 4.8. Flow Cytometric Analyses 

The A2058 and BLM melanoma cells were seeded 1 × 10^6^ in Petri dishes with complete growth medium and cultivated for 24 h. After cell cultivation, the A2058 cells were treated with chalcone **1C** at 10 μmol/L concentration, BLM cells at 20 μmol/L concentration and both melanoma cell lines were treated with DMSO as the negative control at the same concentrations for 24, 48 and 72 h. Adherent and floating cells were harvested, pelleted by centrifugation at 1200 rpm for 5 min. Pellets were resuspended in washing buffer (PBS) and divided for a particular analysis. Afterwards, cells were prepared according to assay kit protocols or fixed with cold 4% paraformaldehyde (15 min) and permeabilized with 90% methanol (10 min on ice) with washing steps (PBS) and stained prior to analysis (Table 5) for 15 min in the dark at room temperature. Fluorescence was detected using a BD FACSCalibur^TM^ Flow Cytometer (Becton Dickinson, San Jose, CA, USA).

### 4.9. Western Blot Analyses

Melanoma cells (A2058 and BLM) were treated with the tested chalcone **1C** (10 μM and 20 μM) for 24, 48 and 72 h. Protein lysates from melanoma cells were prepared using a Laemmle lysis buffer containing glycerol, 1M Tris/HCl (pH 6.8), 20% sodium dodecyl sulfate (SDS), deionized H20, phosphatase and protease inhibitors (Sigma-Aldrich) and a sonication process. The protein concentrations were determined by the Pierce^®^ BCA Protein Assay Kit (Thermo Scientific, Rockford, IL, USA) and measured by an automated Cytation™ 3 Cell Imaging Multi-Mode Reader (Biotek) at a wavelength of 570 nm. Proteins (25–40 μg of sample per well) were separated on SDS-PAA gel (12%) at 100 V for 3 h and transferred to a polyvinylidene difluoride (PVDF) membrane using the iBlot™ 2 Dry Blotting System (Invitrogen, Carlsbad, CA, USA). Membranes with transferred proteins were blocked in 5% BSA (bovine serum albumin; SERVA, Heidelberg, Germany) or 5% non-fat dry milk (Cell Signaling Technology^®^, Danvers, MA, USA) in TBS-Tween (pH 7.4) for 1 h at room temperature to minimise non-specific binding. Blocking was followed by incubation with primary antibodies (Table 6) overnight at 4 °C. The next day, membranes were washed in TBS-Tween (3 × 5 min) and incubated with the corresponding horseradish peroxidase (HRP)-conjugated anti-mouse or anti-rabbit secondary antibody for 1 h at room temperature. After incubation, membranes were washed again in TBS-Tween (3 × 5 min) and the expression of proteins was detected using the MF-ChemiBIS 2.0 Imaging System (DNR BIO-Imaging Systems, Jerusalem, Israel) with chemiluminescent ECL substrate (Thermo Scientific, Rockford, IL, USA). A densitometric analysis of Western Blot (WB) results was performed using the Image Studio™ Lite Software (LI-COR Biosciences, Lincoln, NE, USA). Equal loading was verified by using the β-actin antibody. Three independent analyses were performed.

### 4.10. Statistical Analyses

Results are expressed as mean ± standard deviation (SD). Statistical analyses of the data were performed using standard procedures with one-way analysis of variance (ANOVA) followed by the Bonferroni multiple comparison test. Differences were considered significant when *p* < 0.05. Throughout this paper * indicates *p* < 0.05, ** *p* < 0.01, *** *p* < 0.001 versus vehicle (DMSO).

## 5. Conclusions

Overall, our results demonstrate that the chalcone-acridine hybrid **1C** is a strong suppressor of melanoma cell survival. It induced G2/M cell cycle arrest by modulation of the p53, p21, cyclin B1, and ChK1 expression. Moreover, we also observed that chalcone **1C** promoted apoptosis by disruption of mitochondrial functions as proved by the decrease of MMP, the modulation of the Bcl-2 protein family functions and cytochrome c release with subsequent caspase activation. Furthermore, the activation of several MAP kinases may also have an important role in the antiproliferative and pro-apoptotic effect of this chalcone. Taken together, the presented results showed that this chalcone-acridine hybrid may be a promising agent for melanoma treatment.

## Figures and Tables

**Figure 1 ijms-23-12266-f001:**
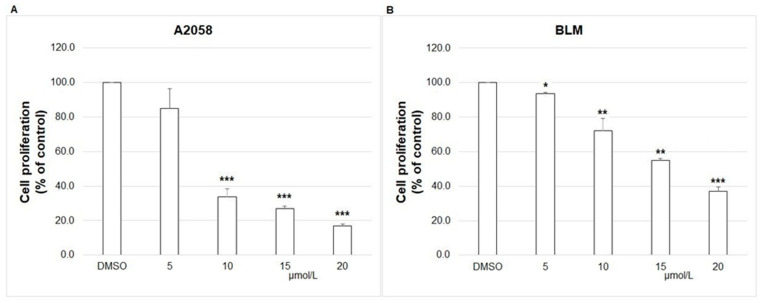
Effect of **1C** on BrdU incorporation in A2058 (**A**) and BLM (**B**) cells. Melanoma cells were exposed to chalcone **1C** at concentrations ranging from 5 to 20 μmol/L for 72 h. The data show the mean ± SD values of three independent experiments. Statistical significance: * *p* < 0.05, ** *p* < 0.01, *** *p* < 0.001 vs. DMSO.

**Figure 2 ijms-23-12266-f002:**
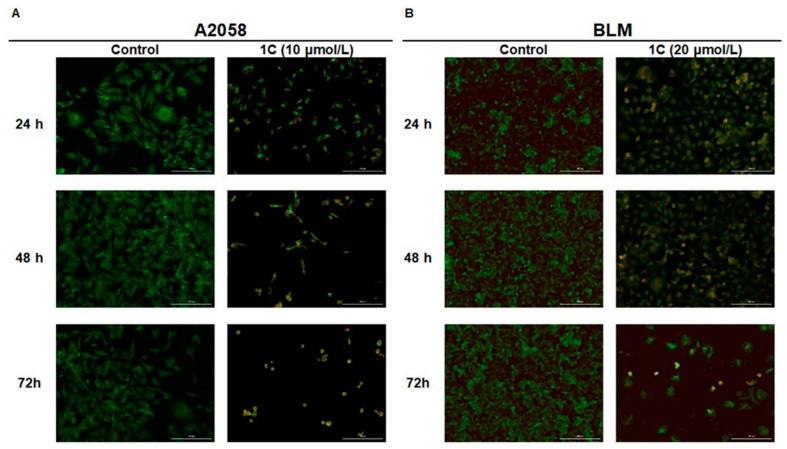
Fluorescence microscopy detection of apoptosis using AO/PI staining in A2058 (**A**) and BLM (**B**) melanoma cells after treatment with chalcone **1C** at a concentration of 10 μmol/L (A2058) and 20 μmol/L (BLM) for 24, 48 and 72 h. Green = living cells, yellow = early apoptotic cells, orange = late apoptotic cells, red = dead/necrotic cells. A representative figure of three independent experiments is presented. Magnification is 100×.

**Figure 3 ijms-23-12266-f003:**
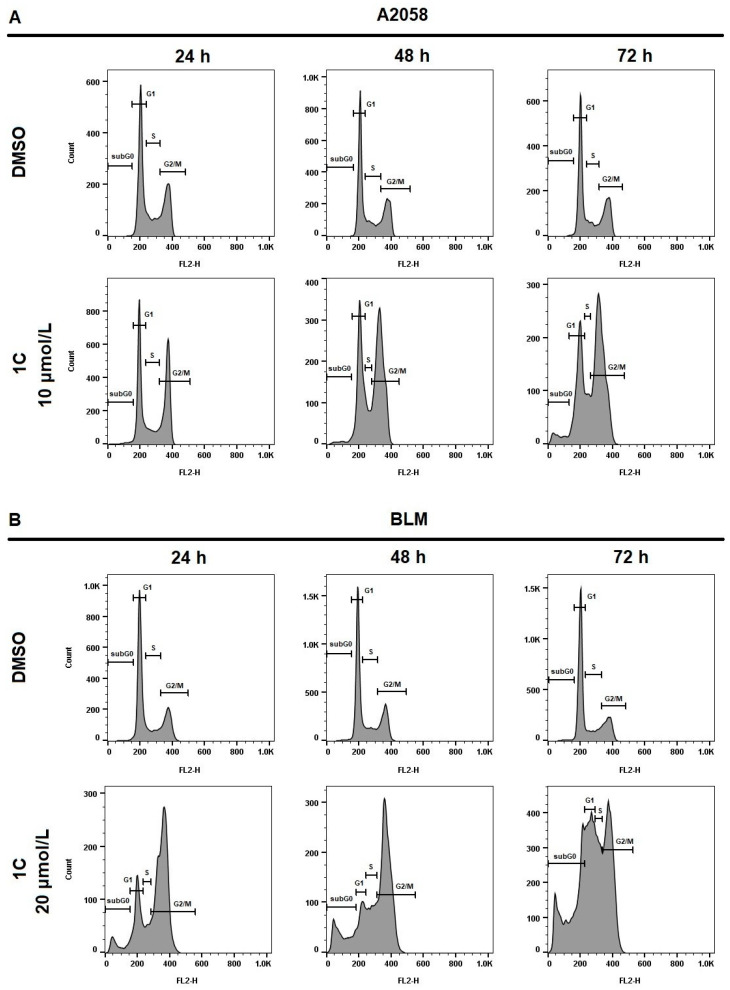
Cell cycle analysis of A2058 (**A**) and BLM (**B**) melanoma cells after treatment with chalcone **1C** for 24, 48 and 72 h at a concentration of 10 μmol/L (A2058) and 20 μmol/L (BLM). Representative histograms from three independent experiments.

**Figure 4 ijms-23-12266-f004:**
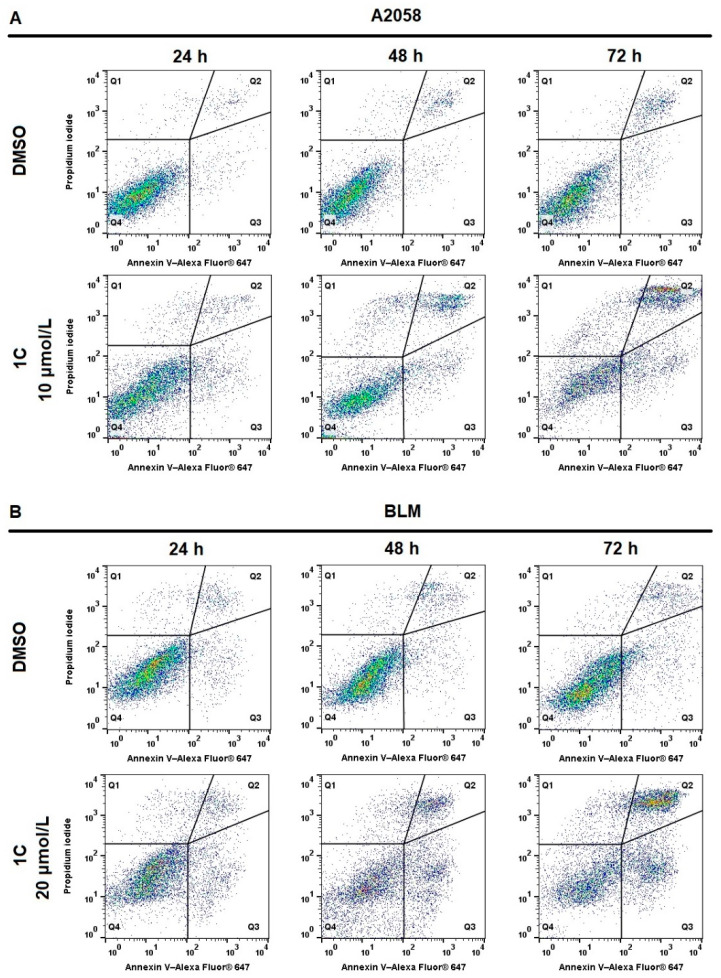
Flow cytometric analysis of **1C**-induced apoptosis after 24, 48 and 72 h in (**A**) A2058 (10 μmol/L) and (**B**) BLM (20 μmol/L) melanoma cells using Annexin V/PI staining. Representative dot plots from three independent experiments.

**Figure 5 ijms-23-12266-f005:**
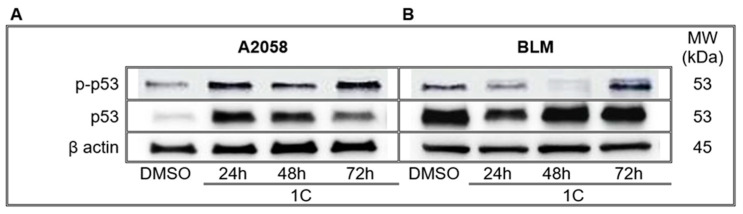
WB analysis of p53 protein in A2058 (**A**) and BLM (**B**) melanoma cells after 24, 48 and 72 h of treatment with chalcone **1C** at concentrations of 10 μmol/L (A2058) and 20 μmol/L (BLM). Representative WB.

**Figure 6 ijms-23-12266-f006:**
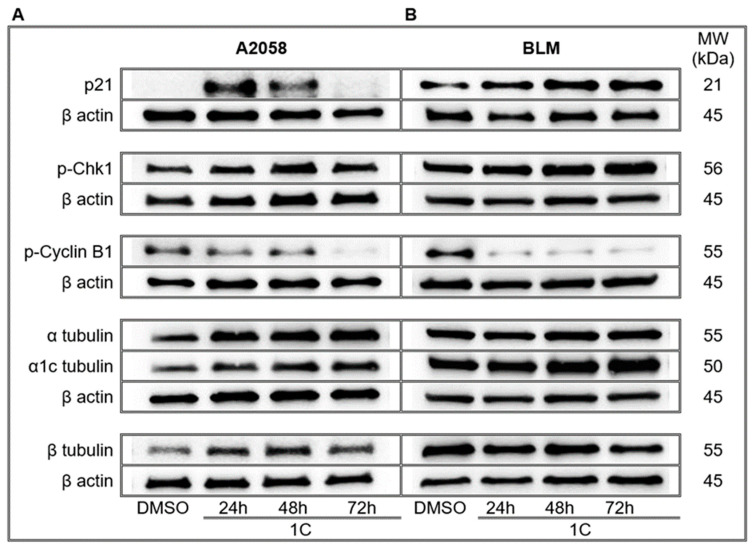
Western blot analysis of cell cycle regulating proteins in A2058 (**A**) and BLM (**B**) melanoma cells after treatment with chalcone **1C** for 24, 48 and 72 h. Representative WB.

**Figure 7 ijms-23-12266-f007:**
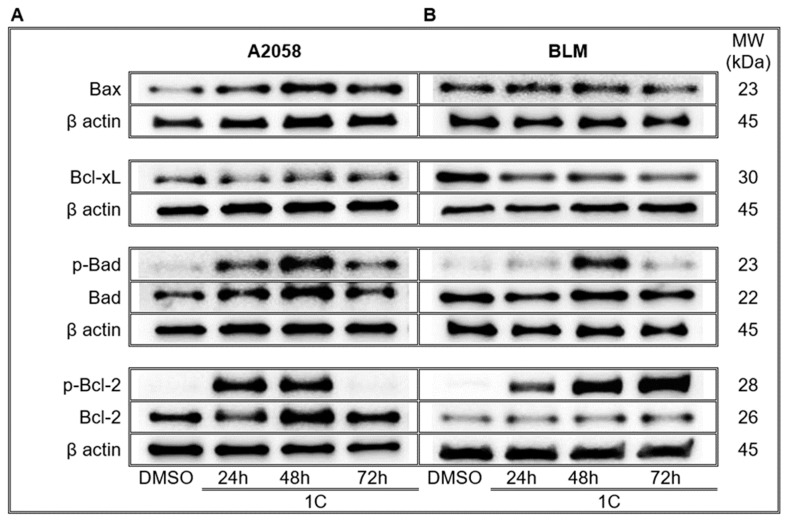
Western blot analysis of mitochondrial apoptosis-related proteins in A2058 (**A**) and BLM (**B**) melanoma cells after treatment with chalcone **1C** for 24, 48 and 72 h. Representative WB.

**Figure 8 ijms-23-12266-f008:**
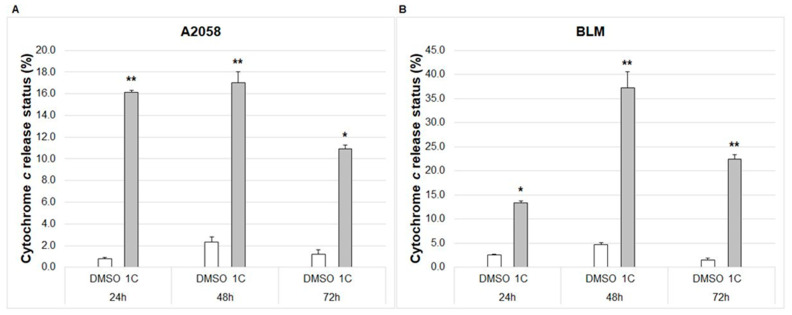
Flow cytometric analysis of released cytochrome c in A2058 (**A**) and BLM (**B**) melanoma cells after 24, 48 and 72 h of treatment with chalcone **1C** at concentrations of 10 μmol/L (A2058) and 20 μmol/L (BLM). The data show the mean ± SD values of three independent experiments. Statistical significance: * *p* < 0.05, ** *p* < 0.01 vs. DMSO (vehicle).

**Figure 9 ijms-23-12266-f009:**
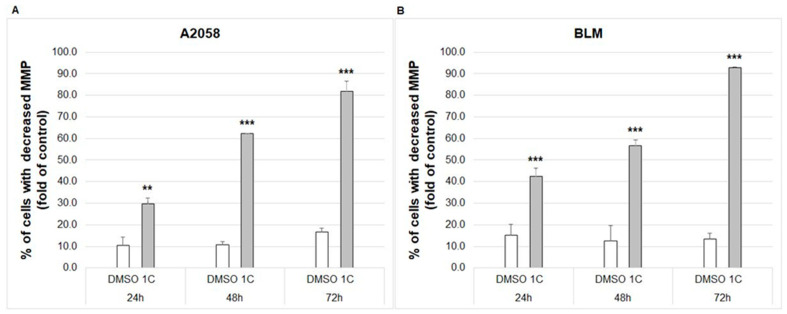
Flow cytometric analysis of changes in mitochondrial membrane potential (MMP) in A2058 (**A**) and BLM (**B**) melanoma cells after 24, 48 and 72 h of treatment with chalcone **1C** at concentrations of 10 μmol/L (A2058) and 20 μmol/L (BLM). The data show the mean ± SD values of three independent experiments. Statistical significance: ** *p* < 0.01, *** *p* < 0.001 vs. DMSO (vehicle).

**Figure 10 ijms-23-12266-f010:**
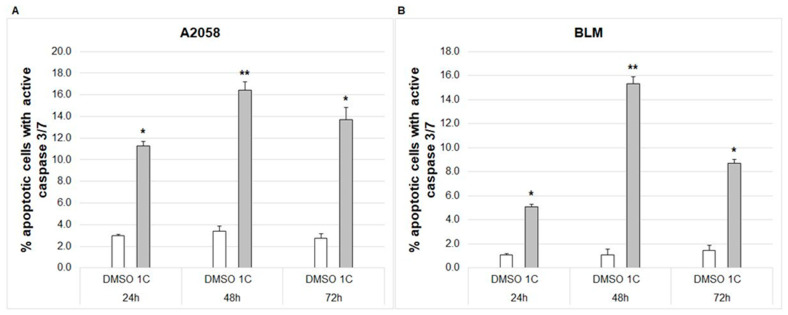
Flow cytometric analysis of caspase-3/7 activity in A2058 (**A**) and BLM (**B**) melanoma cells after 24, 48 and 72 h of treatment with chalcone **1C** at concentrations of 10 μmol/L (A2058) and 20 μmol/L (BLM). The data show the mean ± SD values of three independent experiments. Statistical significance: * *p* < 0.05, ** *p* < 0.01 vs. DMSO (vehicle).

**Figure 11 ijms-23-12266-f011:**
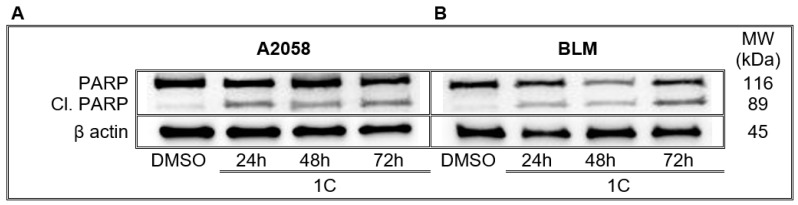
Induction of PARP cleavage in A2058 (**A**) and BLM (**B**) melanoma cells after 24, 48 and 72 h of treatment with chalcone **1C** at concentrations of 10 μmol/L (A2058) and 20 μmol/L (BLM). Representative WB.

**Figure 12 ijms-23-12266-f012:**
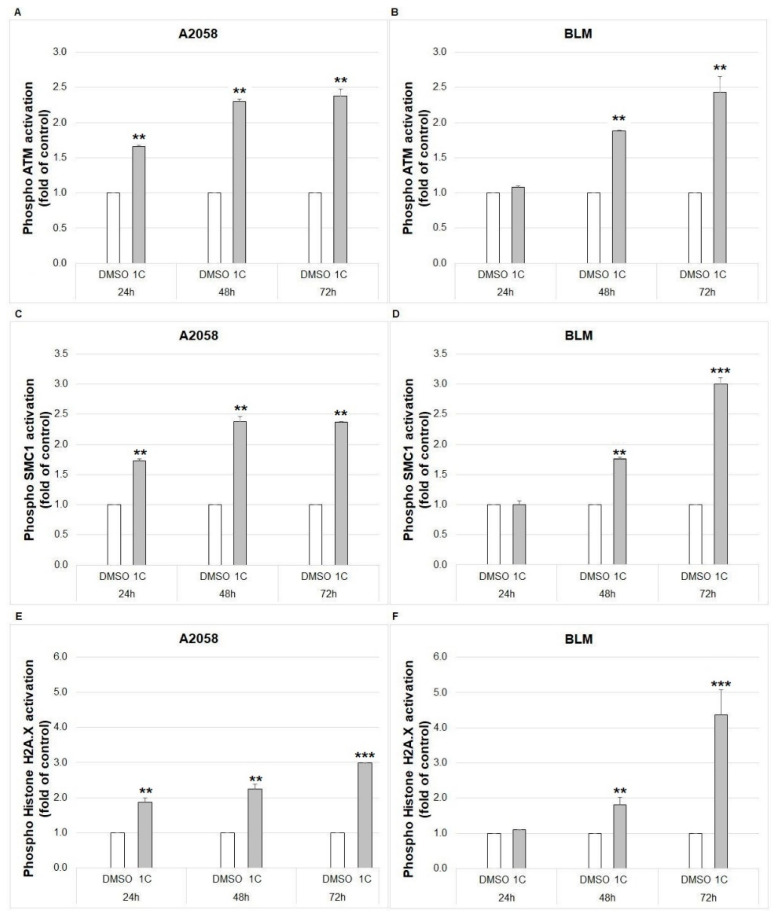
Flow cytometric analysis of DNA damage. Phosphorylation of ATM (**A**,**B**), SMC1 (**C**,**D**) and H2A.X (**E**,**F**) in A2058 and BLM melanoma cells after 24, 48 and 72 h of treatment with chalcone **1C** at concentrations of 10 μmol/L (A2058) and 20 μmol/L (BLM). The data show the mean ± SD values of three independent experiments. Statistical significance: ** *p* < 0.01, *** *p* < 0.001 vs. DMSO (vehicle).

**Figure 13 ijms-23-12266-f013:**
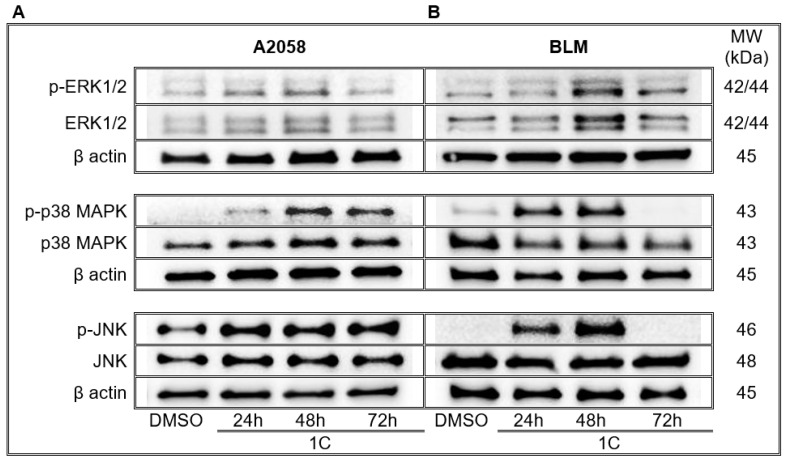
Western blot analysis of ERK1/2, JNK and p38 MAPK signaling pathways in A2058 (**A**) and BLM (**B**) melanoma cells after treatment with chalcone **1C** for 24, 48 and 72 h. Representative WB.

**Figure 14 ijms-23-12266-f014:**
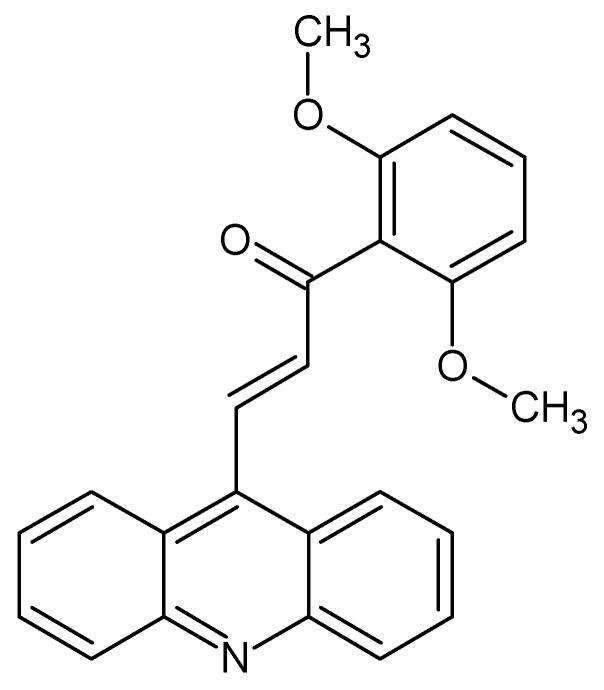
Chalcone **1C** structure ((2E)-3-(acridin-9-yl)-1-(2,6-dimethoxyphenyl)prop-2-en-1-one).

**Table 1 ijms-23-12266-t001:** IC50 (μmol/L) of tested compounds in different cell lines after 72 h incubation.

Compound	Cell Lines		
	A2058	BLM	MCF-10A
**1C**	7.96 ± 0.38	17.93 ± 0.87	36.54 ± 0.87
ZKCH-11A	31.90 ± 0.95	35.27 ± 6.10	85.01 ± 2.19
ZKCH-11C	41.64 ± 0.73	38.55 ± 0.06	>100
ZKCH-11E	37.34 ± 2.28	33.75 ± 2.14	44.71 ± 4.96
ZKCH-11F	45.82 ± 2.54	47.69 ± 1.49	>100
ZKCH-11G	39.96 ± 1.37	39.45 ± 3.90	51.27 ± 3.94
ZKCH-11H	42.43 ± 2.98	40.85 ± 2.62	60.68 ± 3.09

Results are presented as the mean ± SD of three independent experiments.

**Table 2 ijms-23-12266-t002:** Comparison of MTT and BrdU IC50 (μmol/L) of **1C** on melanoma cell lines and noncancer cell line MCF-10A.

Compound	Assay	Cell Lines		
		A2058	BLM	MCF-10A
**1C**	MTT	7.96 ± 0.38	17.93 ± 0.87	36.54 ± 0.87
	BrdU	8.40 ± 0.05	16.51 ± 0.01	32.86 ± 1.56
	Selectivity index	4.6/3.9	2.0/1.99	

Results are presented as the mean ± SD of three independent experiments. The selectivity index was calculated on the basis of MTT or BrdU results.

**Table 3 ijms-23-12266-t003:** Cell cycle analysis of A2058 and BLM melanoma cells after treatment with chalcone **1C** for 24, 48 and 72 h at a concentration of 10 μmol/L (A2058) and 20 μmol/L (BLM).

A2058					
Time	Treatment	subG0	G1	S	G2/M
24 h	DMSO	0.55 ± 0.05	50.75 ± 0.04	16.85 ± 0.61	31.85 ± 0.53
	**1C**	1.30 ± 1.11	34.67 ± 4.92 **	11.79 ± 1.51 *	52.23 ± 7.03 **
48 h	DMSO	0.54 ± 0.15	51.40 ± 0.33	17.15 ± 1.43	30.90 ± 1.63
	**1C**	7.54 ± 3.73 *	24.60 ± 11.45 **	8.12 ± 1.15 *	59.73 ± 12.23 **
72 h	DMSO	0.98 ± 0.02	54.20 ± 0.73	13.00 ± 0.65	31.80 ± 0.08
	**1C**	12.12 ± 5.84 *	18.67 ± 8.56 **	8.85 ± 2.70 *	60.37 ± 4.42 **
**BLM**					
**Time**	**Treatment**	**subG0**	**G1**	**S**	**G2/M**
24 h	DMSO	2.43 ± 0.24	56.70 ± 0.90	15.90 ± 0.33	25.00 ± 0.98
	**1C**	8.22 ± 1.62 *	19.35 ± 1.43 **	9.19 ± 0.28 *	63.25 ± 0.45 ***
48 h	DMSO	2.10 ± 0.08	61.60 ± 3.76	14.75 ± 1.18	21.55 ± 2.65
	**1C**	17.05 ± 0.45 *	11.65 ± 0.69 **	16.35 ± 0.20	54.95 ± 0.45 **
72 h	DMSO	1.87 ± 0.02	63.90 ± 2.45	16.40 ± 0.33	17.85 ± 2.08
	**1C**	24.30 ± 3.84 **	13.85 ± 1.18 ***	32.60 ± 2.69 *	29.25 ± 0.04 *

The data show the mean ± SD values of three independent experiments. Statistical significance: * *p* < 0.05, ** *p* < 0.01, *** *p* < 0.001 vs. DMSO (vehicle).

**Table 4 ijms-23-12266-t004:** Apoptosis analysis of A2058 and BLM melanoma cells after treatment with chalcone **1C** for 24, 48 and 72 h at a concentration of 10 μmol/L (A2058) and 20 μmol/L (BLM).

A2058					
Time	Treatment	Live An^−^/PI^−^	Early Apo An^+^/PI^−^	Late Apo An^+^/PI^+^	Dead An^−^/PI^+^
24 h	DMSO	88.60 ± 3.67	5.83 ± 2.04	3.93 ± 1.25	1.65 ± 0.35
	**1C**	84.80 ± 0.24	7.53 ± 1.17	4.22 ± 0.47	3.41 ± 0.46
48 h	DMSO	88.75 ± 0.94	4.69 ± 1.56	4.63 ± 0.61	1.96 ± 0.02
	**1C**	68.70 ± 1.63 **	11.52 ± 0.42 *	13.85 ± 2.57 *	5.94 ± 0.52
72 h	DMSO	84.20 ± 0.65	7.76 ± 0.00	6.88 ± 1.42	1.19 ± 0.06
	**1C**	52.00 ± 5.23 **	13.30 ± 1.71 *	23.05 ± 4.94 **	11.65 ± 1.43 *
**BLM**					
**Time**	**Treatment**	**Live** **An^−^/PI^−^**	**Early Apo An^+^/PI^−^**	**Late Apo** **An^+^/PI^+^**	**Dead** **An^−^/PI^+^**
24 h	DMSO	85.00 ± 1.39	8.60 ± 0.14	3.95 ± 0.60	2.42 ± 0.66
	**1C**	73.90 ± 0.49 *	12.96 ± 0.77 *	6.90 ± 0.09	6.25 ± 0.20
48 h	DMSO	85.95 ± 4.04	7.37 ± 0.96	4.29 ± 1.88	2.41 ± 1.18
	**1C**	61.50 ± 2.61 **	20.25 ± 1.27 *	12.40 ± 0.49 *	5.81 ± 0.86
72 h	DMSO	83.70 ± 1.71	7.95 ± 0.21	6.17 ± 1.29	2.18 ± 0.24
	**1C**	39.60 ± 0.90 **	25.95 ± 3.80 **	28.55 ± 3.23 **	5.90 ± 1.47 *

The data show the mean ± SD values of three independent experiments. Statistical significance: * *p* < 0.05, ** *p* < 0.01 vs. DMSO (vehicle).

**Table 5 ijms-23-12266-t005:** Flow Cytometry Staining.

Analysis	Staining	Manufacturer
Caspase activation	CellEvent™ Caspase-3/7 Green Flow Cytometry Assay Kit	Thermo Scientific, Rockford, IL, USA
Cytochrome *c* release	Cytochrome *c* Antibody (6H2) FITC Conjugate	Invitrogen, Carlsbad, CA, USA
Mitochondrial membrane potential	TMRE (Tetramethylrhodamine ethyl ester perchlorate) final concentration 0.1 μmol/L	Sigma-Aldrich, St. Louis, MO, USA
DNA damage	FlowCellect™ Multi-Color DNA Damage Response Kit	Millipore Corporation, Temecula, CA, USA

**Table 6 ijms-23-12266-t006:** List of Western blot antibodies.

Primary Antibodies	Mr (kDa)	Origin	Dilution	Manufacturer
Bad	22	Rabbit	1:1000	Abcam, Cambridge, UK
Phospho Bad	23	Rabbit	1:1000	Cell Signalling Technology^®^, Danvers, MA, USA
Bcl-2	26	Mouse	1:1000	Abcam, Cambridge, UK
Phospho Bcl-2	28	Rabbit	1:1000	Cell Signalling Technology^®^, Danvers, MA, USA
Bax	23	Mouse	1:1000	Santa Cruz Biotechnology, Inc. Dallas, TX, USA
Bcl-xL	30	Rabbit	1:1000	
α Tubulin	55	Rabbit	1:1000	
α1C Tubulin	50	Mouse	1:1000	
β Tubulin	55	Rabbit	1:1000	
p21	21	Rabbit	1:1000	Cell Signalling Technology^®^, Danvers, MA, USA
Phospho-Cyclin B1	55	Rabbit	1:1000	
Phosho-Chk1	56	Rabbit	1:1000	
p38 MAPK	43	Rabbit	1:1000	
Phospho p38 MAPK	43	Rabbit	1:1000	
p44/42 MAPK (Erk1/2)	42/44	Rabbit	1:1000	
Phospho p44/42 MAPK (Erk1/2)	42/44	Mouse	1:1000	
JNK	48	Mouse	1:1000	Thermo Scientific, Rockford, IL, USA
Phospho SAPK/JNK	46/54	Mouse	1:1000	Cell Signalling Technology^®^, Danvers, MA, USA
PARP	116/89	Rabbit	1:1000	
p53	53	Rabbit	1:1000	
Phospho p53	53	Rabbit	1:1000	
β-actin	45	Mouse	1:2500	
**Secondary Antibodies**				
Anti-mouse IgG HRP	-	Goat	1:1000	Cell Signalling Technology^®^, Danvers, MA, USA
Anti-rabbit IgG HRP	-	Goat	1:1000	

## Data Availability

Not applicable.

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
