# Peer review of "Chalcone-Acridine Hybrid Suppresses Melanoma Cell Progression via G2/M Cell Cycle Arrest, DNA Damage, Apoptosis, and Modulation of MAP Kinases Activity"

_ijms, 2022, doi:10.3390/ijms232012266_

Round 1

Reviewer 1 Report

The Authors presented an interesting and well organized and well-written paper. However, the paper would be much improved if the Authors provided more information on the tested compounds (i.g., structural differences between them, etc.)

Additionally, on what basis did the Authors choose to determine compounds activity at doses of 100, 50 and 10 μmol/L(MTT test)? Where there any additional studies which indicated that micromolar concentration is sufficent to observe any effect?

Since there is no information about the structure of 1c compound, it is difficult to state how this compound may influence such an activity presented in the paper. Nonetheless, I wonder whether the Authors tried to determine which pharmacophore (structure's element) is mostly responsible for this effect. Also, since hybrid compounds are known to exert their effects due to additive or even synergistic interactions between building pharmacophores, can the Authors provide any information how this drug act when compare at least to one of its building element?

Author Response

We would like to thank the reviewer for his/her comments, which have given us the opportunity to improve the manuscript (see attached file).

Reviewer 2 Report

The authors investigated the effect of synthetic chalcones on melanoma cancer cell proliferation and identified one compound with the most potent antiproliferative properties. To assess the mechanism of action, the authors extended the study and included in their analyses assays for the quantification of metabolic activities, cell-cycle progression and apoptosis. The authors observed drug-induced cell-cycle arrest of the cells accompanied by changes in the expression and phosphorylation of cell-cycle proteins, including pathways of DNA damage and MAP-associated signalling.

On the basis of their findings and results from previous work, the authors discuss a potential mechanism, where they propose an involvement of the drug in promoting DNA damage and interfering with pathways affecting DNA repair and apoptosis.

I think this is a nice piece of work. Before I recommend publication, I have the following questions:

1. The authors mention that the cells increasingly detach if they add the drug. This does not mean that the cells are dead or that they went into apoptosis, they could just lose adhesion without dying. The authors should distinguish between viable and non-viable attached and detached cells and show this.   

2. Have the authors considered that a decrease in proliferation as claimed, could just be an effect of less cells on the coverslip, because of detachment or because they divide less?

3. In Figure 5, the authors describe an increase in p-p53 between 24 and 72h, but I cannot see that, it seems to be the same. And, at 48h there is always less phosphorylated p53, why? Also in Fig. 6, there is no p21 after 72h detectable. How does this correlate with higher p53? The authors should clarify these aspects.

Author Response

(The authors gave the same response as above.)

Reviewer 3 Report

The authors try to find out how the chalcone-acridine hybrid works on melanoma cells. The article was written very well, very carefully. The authors analyze several potential mechanisms and places of influence. The article is a continuation of the research of this research team (https://link.springer.com/article/10.1007/s00044-022-02911, https://analyticalsciencejournals.onlinelibrary.wiley.com/doi/10.1002/mrc.5028). I think that with a few corrections the article will be suitable for publication.

Comments and questions:

1) Line 15: "chalcone complexes" I would rather write "chalcone-acridine hybrids"

2) Throughout the article, I recommend that you write "1C" as the abbreviation for a chemical compound in bold.

3) Lines 52-53: Also other chalcones with free amine groups have anti-cancer and anti-microbial activity, e.g. 10.3390/molecules 24224129.

4) What are the selectivity indexes for each line?

5) In section 4.1. the authors give the names of the compounds. I understand that it is not their intention to present a synthesis of them, but I would expect appropriate citations for the synthesis path.

6) I would also recommend putting the compound formula 1C e.g. in section 2.

7) What was the purity of the test compound? What method was used to determine it? Please specify.

8) LInia 430: "1H, 13C" - digits should be superscript

9) Section 4.2. What are the ATCC numbers of cell lines? From which supplier were they purchased, please specify.

10) Lines 445 and 469: digits should be subscript.

11) Lines 461, 473, 486, 501, 511: digits should be superscript.

12) Line 449: Please provide the manufacturer of the compound used for the MTT.

Author Response

(The authors gave the same response as above.)
